# A Novel Electrochemiluminescence Immunosensor Based on Resonance Energy Transfer between g-CN and NU-1000(Zr) for Ultrasensitive Detection of Ochratoxin A in Coffee

**DOI:** 10.3390/foods12040707

**Published:** 2023-02-06

**Authors:** Linzhi Li, Xiaofeng Wang, Jian Chen, Tianzeng Huang, Hongmei Cao, Xing Liu

**Affiliations:** 1School of Food Science and Engineering, Hainan University, Haikou 570228, China; 2School of Environmental Science, Nanjing Xiaozhuang University, Nanjing 211171, China; 3School of Chemical Engineering and Technology, Hainan University, Haikou 570228, China

**Keywords:** Nb28-C4bpα heptamer, electrochemiluminescence immunosensor, resonance energy transfer, ochratoxin A

## Abstract

In this study, an electrochemiluminescence (ECL) immunosensor based on nanobody heptamer and resonance energy transfer (RET) between g-C_3_N_4_ (g-CN) and NU-1000(Zr) was proposed for ultrasensitive ochratoxin A (OTA) detection. First, OTA heptamer fusion protein was prepared by fusing OTA-specific nanometric (Nb28) with a c-terminal fragment of C4 binding protein (C4bpα) (Nb28-C4bpα). Then, Nb28-C4bpα heptamer with the high affinity used as a molecular recognition probe, of which plenty of binding sites were provided for OTA-Apt-NU-1000(Zr) nanocomposites, thereby improving the immunosensors’ sensitivity. In addition, the quantitative analysis of OTA can be achieved by using the signal quenching effect of NU-1000(Zr) on g-CN. As the concentration of OTA increases, the amount of OTA-Apt-NU-1000(Zr) fixed on the electrode surface decreases. RET between g-CN and NU-1000(Zr) is weakened leading to the increase of ECL signal. Thus, OTA content is indirectly proportional to ECL intensity. Based on the above principle, an ultra-sensitive and specific ECL immunosensor for OTA detection was constructed by using heptamer technology and RET between two nanomaterials, with a range from 0.1 pg/mL to 500 ng/mL, and the detection limit of only 33 fg/mL. The prepared ECL-RET immunosensor showed good performance and can be successfully used for the determination of OTA content in real coffee samples, suggesting that the nanobody polymerization strategy and the RET effect between NU-1000(Zr) and g-CN can provide an alternative for improving the sensitivity of important mycotoxin detection.

## 1. Introduction

Ochratoxin A (OTA) is one of the significant mycotoxins, a toxic secondary metabolite produced by Aspergillus and Penicillium, which is widely distributed in nature [1]. It is easy to contaminate many types of foods, including grains, coffee beans, and spices. Furthermore, OTA with the stable chemical properties is not easily degraded quickly, making it difficult to eliminate once it enters the body through food [2]. Long-term OTA accumulation will cause irreversible harm to humans and animals, leading to deformities and cancer [3,4]. Today, OTA is a major mycotoxin that International Agency for Research on Cancer (IARC) has designated as a human IIB carcinogen. To reduce the risk of OTA to human health, organizations and countries have established the maximum limit of OTA in food [5].

Coffee is a significant cash crop in China, particularly in Hainan province. However, due to Hainan’s warm climate, the coffee is susceptible to OTA contamination. As a result, coffee pollution has caused widespread concern among researchers [6,7]. In our country, the maximum OTA limit for coffee beans or coffee powder is 5.0 µg/kg, and the maximum OTA limit for instant coffee is 10.0 µg/kg (GB 5009.96-2016). To reduce OTA exposure, it is critical to develop sensitive and accurate analytical techniques for OTA detection in coffee. To date, high-performance liquid chromatography (HPLC) [8], fluorescence resonance energy transfer on lateral flow immunoassay (FRET-LFI) [9], and enzyme-linked immunosorbent assay (ELISA) [10,11] are used to detect OTA in coffee. Undoubtedly, these techniques can provide a high sensitivity, however they have some drawbacks, such as an expensive analyzer, the high cost, and a high interference of the sample matrix. Electrochemiluminescence (ECL) is a potential alternative for detecting ochratoxins due to its high sensitivity, stability, and low background signal [12,13,14,15]. Resonance energy transfer (RET) based ECL was developed for a highly specific and susceptible ECL biosensing system which is attributed to further improving ECL response, which occurs due to the large area overlap between the spectra of the energy donor and acceptor, as well as the close distance (<10 nm) between them [16,17,18]. Using RET in an ECL assay can significantly improve its accuracy and sensitivity. In recent years, researchers have been working hard to create suitable ECL-RET donor-acceptor pairs as well as a variety of ECL-RET sensing platforms for quantitative OTA analysis. For example, Wei et al. built a simple and sensitive ECL aptasensor for OTA determination, inspired by the benefits of DNA walking machines and the properties of cadmium sulfide semiconductor quantum dots (CdS QDs) [19]. Gao et al. fabricated an innovative aptasensor based on ECL-RET from CdTe QDs to a cyanine dye (Cy5) fluorophore to determine OTA [20]. However, in the current study, energy donors of ECL-RET sensor was primarily by QDs or luminescent reagents. Therefore, in this work, it is a new attempt to directly use two nanomaterials as donor-acceptor pairs to propose ECL-RET sensor for OTA analysis.

Graphite-phase C_3_N_4_ nanosheets (g-C_3_N_4_, referred to as g-CN), have been widely used in the field of photochemistry and electrochemistry due to their excellent photocatalytic performance and ECL activity [21,22,23,24]. Metal-organic framework (MOF) materials with good chemical stability and large specific surface area are often used as fixed carriers for detection, identification, and sensor platform construction [25,26,27,28]. In this study, we successfully proposed a new ECL-RET sensor with g-CN as energy donor and NU-1000(Zr) as an energy acceptor for OTA detection. ECL-RET between g-CN and NU-1000(Zr) opens up new directions for materials applications. The constructed ECL-RET immunosensor is not only sensitive, but also safer and more environmentally friendly without the use of additional luminescent reagents.

Currently, nanobodies (Nbs) have attracted wide attention. Compared with traditional antibodies, Nbs show excellent the advantages, such as small molecular weight, easy expression, and high affinity, and are more suitable for maturation and polymerization in vitro [29]. Nbs polymerization usually has a longer half-life, stronger stability, and higher affinity than Nbs [30]. Nb28-C4bpα heptamer were constructed using C4 binding protein (C4bpα) as an excellent infrastructure, with high affinity and stability compared to monomers [11]. Furthermore, ELISA assay has performed using Nb28-C4bpα heptamer, significantly increasing the sensitivity of the assay [11]. However, few research was conducted to enhance ECL immunosensing sensitivity using nanobody polymeric as biometric probes. Therefore, we attempted to construct ECL-RET immunosensor using Nb28-C4bpα heptamer and OTA aptamer with high affinity for OTA molecules as recognition probes.

Herein, we proposed a novel ECL immunosensor based on RET between g-CN and NU-1000(Zr) for sensitive OTA detection. In this work, first, rod-like NU-1000(Zr) nanoparticles with the high surface area were synthesized. Then, they can bind large amounts of OTA molecules through the interaction between OTA and OTA aptamer as recognition element (defined as OTA-Apt-NU-1000(Zr)). To further improve ECL sensitivity, OTA Nbs polymer Nb28-C4bpα was developed and applied to ECL-RET sensor as recognition elements. In the test, OTA in the sample binds to Nb28-C4bpα on the surface of the ECL-RET immunosensor, and OTA-Apt-NU-1000(Zr) combined with its remaining sites. As OTA concentration increases, the amount of OTA-Apt-NU-1000(Zr) fixed on the electrode surface decreases. Therefore, RET between g-CN and NU-1000(Zr) is attenuated, leading to an increase in ECL response. Based on this principle, the quantitative analysis of OTA can be achieved by the signal quenching effect. Compared with the ordinary “turn off” sensor, the fabricated “turn on” ECL-RET immunosensor shows more sensitive signal response to the target. Thus, the developed ECL-RET immunosensor displays a wide linear range and low detection limit, providing a new strategy for the detection of mycotoxins in food.

## 2. Materials and Methods

### 2.1. Chemical Reagents and Apparatus

Absolute ethyl alcohol, benzoic acid, HNO_3_, KCl, K_3_Fe(CN)_6_, K_4_Fe(CN)_6_·3H_2_O, and K_2_S_2_O_8_ were purchased from Aladdin Co. Ltd. (Shanghai, China). Lysozyme, Nickel-nitrilotriacetic acid (Ni-NTA), Sepharose, Phenylmethyl sulfonyl fluoride (PMSF), and color mixed protein marker PR1920 (11–245KD) were purchased from Solarbio Science & Technology Co., Ltd. (Beijing, China). N, N-Dimethylformamide (DMF), dichloromethane, 1,3,6,8-Tetra(4-carboxyphenyl)pyrene (H_4_PTPA), and Zirconyl chloride octahydrate (ZrOCl_2_·8H_2_O) were procured from Macklin Biochemical Co., Ltd. (Shanghai, China). Bovine serum albumin (BSA), N-Hydroxy succinimide (NHS), and 1-(3-Dimethylaminopropyl)-3-ethylcarbodiimide hydrochloride (EDC) were ordered from Sigma Life Science Co. Ltd. (Shanghai, China). g-CN was received from Nanjing XFNANO Technology Co., Ltd. (Nanjing, China). OTA-Apt was synthesized from Sangon Biotech Co., Ltd. (Shanghai, China), the OTA aptamer with -NH_2_ modification (5′-NH_2_GATCGGGTG TGGGTGGCGTAAAGGGAGCATCGGACA-3′). Ochratoxin B (OTB), ochratoxin C (OTC), aflatoxin B1 (AFB1), fumonisins B1 (FB1), zearalenone (ZEN), and deoxynivalenol (DON) were obtained from Pribolab Co., Ltd. (Qingdao, China). The coffee sample was obtained from the supermarket in Haikou. All solutions were prepared by deionized water obtained from the Millipore water purification system (18.2 MΩ cm^−1^, Milli-Q). Other reagents were analytical grade without further purification. The size and morphologies of the g-CN and NU-1000(Zr) nanocomposites were recorded by scanning electron microscope (SEM, FEI Helios G4 CX) and transmission electron microscopy (TEM, FEI Talos F200s). The cyclic voltammetry (CV) experiments were carried out on the CHI660e electrochemistry workstation (Shanghai CH Instruments, Shanghai, China) in 0.1 M KCl containing 5 mM [Fe(CN)_6_]^3−^ and the ECL signals were taken by a model MPI-E electrochemiluminescence analyzer (Xi’An Remax Electronic Science & Technology Co. Ltd., Xi’an, China) in 0.1 M PBS containing 20 mM K_2_S_2_O_8_ and 0.1 M KCl. The standard three-electrode system was used in the experiment, with the modified glass carbon electrode (GCE) as the working electrode, the platinum wire as the auxiliary electrode, and the saturated calomel electrode (SCE) as the reference electrode for CV, the Ag/AgCl (3.5 M KCl) as the reference electrode for ECL.

### 2.2. Carboxylation and Activation of g-CN

The carboxylation of g-CN was carried out using the method reported in the literature [31]. First, 1 g g-CN powder was placed in 100 mL 5 M HNO_3_ and refluxed at 125 °C for 24 h. After natural cooling to 25 °C, the refluxed products were centrifuged. Then, products were washed with ultrapure water. Carboxylated g-CN was obtained 12 h after being vacuum-dried at 35 °C. Then, a mixture of 1.5 mL 0.4 M EDC and 0.1 M NHS was added to 3 mg g-CN and shaken for 6 h on a 200 rpm on a thermostatic shaker before being washed and centrifuged 3 times. The activated g-CN was re-dispersed in 1.5 mL 0.001% chitosan acetic acid solution and stored at 4 °C for further use.

### 2.3. The Preparation of NU-1000(Zr) and OTA-Apt-NU-1000(Zr)

NU-1000(Zr) is prepared according to other work [32,33]. H_4_PTPA (34 mg, 0.05 mmol), ZrOCl_2_·8H_2_O (120 mg, 0.36 mmol), and benzoic acid (1.2 g, 10 mmol) were dissolved in 50 mL of DMF and stirred (300 rpm) at 90 °C for 12 h. After the reaction was done, NU-1000(Zr) nanoparticles were centrifugated (10,000 rpm, 15 min) and washed with 15 mL fresh DMF, dichloromethane, and ethanol, respectively. For further characterization and analysis, the resulting NU-1000(Zr) nanoparticles were re-suspended in DMF. For the preparation of OTA-Apt-NU-1000(Zr), firstly, 1.5 mL 0.4 M EDC and 0.1 M NHS mixed solution was added into 3 mg NU-1000(Zr), and the mixture was shaken at 200 rpm for 6 h in a constant temperature shaker, then washed and centrifuged 3 times to obtain activated NU-1000(Zr). Then, 1.5 mL 0.01 M PBS solution was added into 3 mg activated NU-1000(Zr) and 1 OD OTA aptamer (OTA-Apt), and the solution was incubated at 200 rpm for 12 h on a constant temperature shaker, processed by centrifugation, and washed 3 times with ultrapure water to obtain Apt-NU-1000(Zr). Finally, 3 mg Apt-NU-1000(Zr) was mixed with 1 mL OTA (10 mg/mL) and shaken at 200 rpm for 4 h. After washing 3 times, the prepared OTA-Apt-NU-1000(Zr) was re-dispersed in 1.5 mL 0.01 M PBS.

### 2.4. Expression, Purification, and Identification of Nb28-C4bpα Fusion Proteins

As previously reported, we expressed Nb28-C4bpα fusion protein [11]. To obtain the fusion proteins, the *E. coli* Rosetta chemically competent cells containing the vector pET25b-Nb28-C4bpα was used for auto-induction. First, the strain was inoculated in LB medium (containing 100 µg/mL of ampicillin) and then incubated overnight at 37 °C with shaking at 250 rpm until OD_600_ reached 0.5–0.7. Bacterial cells were collected by centrifugation after Nb28-C4bpα was expressed in 25 °C conditions with intense shaking overnight. The cells were then resuspended in a 20 mL equilibration buffer (1 mM PMSF, 8 mM Na_2_HPO_4_, 2 mM KH_2_PO_4_, 2.6 mM KCl, 136 mM NaCl, and 60 mg lysozyme). The resuspended E. coli cells were subjected to ultrasound in an ice bath to prevent degradation or denaturation of the target proteins. Soluble Nb28-C4bpα fusion proteins were obtained by centrifugation (8000× *g*, 4 °C) and filtered through a syringe filter with a 0.22 µm pore size. Finally, Ni-NTA Sepharose and PBS were used to purify and perform dialysis fusion on proteins. The purity and concentration of fusion protein were analyzed by sodium dodecyl sulfate-polyacrylamide gel electrophoresis (SDS-PAGE) and microdroplet ultramicro spectrophotometer, respectively.

### 2.5. Fabrication Process of ECL-RET Immunosensor between g-CN and NU-1000(Zr)

Figure 1 describes the assembly and recognition process of the fabricated ECL-RET immunosensor. First, glassy carbon electrodes (GCE) were polished and cleaned to obtain a bright mirror surface. Then, 5 µL activated g-CN (2 mg/mL) nanosheets were added dropwise to the surface of GCE and dried naturally. To capture the target molecule, 5 µL Nb28-C4bpα heptamer solution was coated, and incubated at 37 °C for 1 h. Then, the modified electrode was washed with ultrapure water to remove the unbound Nb28-C4bpα heptamer molecule. During this process, the amino group of Nb28-C4bpα forms a covalent bond with the carboxylated g-CN. Nb28-C4bpα/g-CN/GCE was then incubated with BSA solution (5 µL, 1%) at 37 °C for 0.5 h to block the non-specific binding sites. After careful cleaning, the obtained BSA/Nb28-C4bpα/g-CN/GCE modified electrode incubated in OTA concentrations of 37 °C for 2 h, then gently washed with distilled water and dried naturally. Finally, 5 µL OTA-Apt-NU-1000(Zr) was dripped onto the modified electrode and incubated at 37 °C for 1 h, and then rinsed with ultrapure water. The OTA-Apt-NU-1000(Zr)/OTA/BSA/Nb28-C4bpα/g-CN/GCE ECL-RET immunosensor was stored at 4 °C for future use.

## 3. Results

### 3.1. Characterizations of g-CN and NU-1000(Zr)

g-CN nanomaterial provides an ECL signal for the immunosensor, and its morphology has an essential influence on the performance of the ECL sensor. The morphology of synthesized g-CN was characterized by SEM. As seen from Figure 1A,B that g-CN nanomaterial has a lamellar structure with a large number of pores between the lamellar layers, which is similar to those reported literatures [34,35]. This structure effectively increases the contact area between g-CN and the S_2_O_8_^2−^ solution, resulting in an obvious ECL signal. In addition, EDS and mapping analysis show that only C and N elements were present in g-CN and evenly distributed. As for NU-1000(Zr), its morphology was characterized by SEM (Figure 1A,B) and TEM (Figure 1C–E).

As can be seen from Figure 2, NU-1000(Zr) is a uniform square rod structure with a length of about 1 µm and a thickness of 150 nm. It can be seen that C, N, and Zr elements are uniformly distributed in NU-1000(Zr) in EDS and mapping.

The elongated rod structure provides a large number of binding sites for OTA-Apt, thereby immobilizing more OTA molecules. When a large amount of NU-1000(Zr) are fixed on the surface of the ECL immunosensor, the ECL-RET efficiency between g-CN and NU-1000(Zr) is enhanced, leading to a significant change in the ECL signal.

### 3.2. Expression, Purification, and Identification of Nb28-C4bpα

Nb28-C4bpα fusion protein was developed to improve the sensitivity of Nb28-based immunoassay. Purification and polymerization of Nb28-C4bpα fusion protein were analyzed by SDS-PAGE. By running SDS-PAGE under non-reduction conditions, two clear target bands with definite monomers and heptamers were observed.

SDS-PAGE was used to characterize the purified Nb28 monomer and Nb28-C4bpα heptamer. Figure 3 shows that both Nb28 monomer and Nb28-C4bpα heptamer have only one band, suggesting high antibody purity. SDS-PAGE also detected an obvious target protein band of Nb28 monomer with a molecular weight of 30 kDa approximately. The band length of the Nb28-C4bpα heptamer target protein exceeds 210 kDa. The Nb28-C4bpα fusion protein was proved to be assembled into heptamer by intermolecular disulfide bonds.

### 3.3. Feasibility Analysis of RET between g-CN and NU-1000(Zr)

The fluorescence emission spectrum (FL emission spectrum) of g-CN and UV-visible absorption spectrum (UV-vis) of NU-1000(Zr) were used to verify the existence of effective ECL-RET between the two nanomaterials. Figure 4 illustrates a strong FL emission peak of g-CN at 455 nm, the FL spectrum of g-CN overlaps with the UV-vis of NU-1000(Zr) at 400 nm–500 nm. According to the literature, the ECL emission spectrum of g-CN has a strong ECL emission peak at 400–600 nm, which overlaps almost completely with the FL spectrum, which is feasible in principle [24,36,37].

### 3.4. Electrochemical and ECL Behaviors of the ECL-RET Immunosensor

The construction process of the modified electrode in 5 mM K_3_Fe(CN)_6_ containing 0.1 M KCl was characterized by CV curves. As exhibited in Figure 5a, bare GCE showed a quasi-reversible redox peak in the presence of a redox probe (curve a). After modification of g-CN on the GCE surface (curve b), there was little effect on electron transfer, and the change of redox current can be ignored. When Nb28-C4bpα heptamer was immobilized on the modified electrode (curve c), the peak current was markedly decreased, which may be attributed to the insulation barrier generated by Nb28-C4bpα heptamer and its high resistance to electron transfer at the electrode/electrolyte interface. After using BSA to block the non-specific active site of the above electrode (curve d), the peak current is reduced due to the blocking effect of this protein on interfacial electron transfer. When OTA was incubated on the modified electrode (curve e), the peak current of CV continued to decrease due to the blocking effect on the redox probe (curve e). After OTA-Apt-NU-1000(Zr) was assembled to the modified electrode, the peak current decreased significantly (curve f) due to the fact that the nanomaterials with lower conductivity have a great blocking effect on the redox probe. The result of the CV signal demonstrates that the modified electrode is successfully constructed.

To further verify the quenching effect of NU-1000(Zr) on the signal of g-CN by ECL-RET, the ECL response of different types of modified electrodes was measured in the detection solution. The reaction mechanism of ECL-RET is based on a g-CN-K_2_S_2_O_8_ system as follows. In this reaction system, firstly, g-CN^−^, SO_4_^2−^, and SO_4_^−^ are generated from g-CN and S_2_O_8_^2−^, respectively, and g-CN^−^ reacts with SO_4_^−^ to form the excited state g-CN^*^. The excited state g-CN^*^(g-C_3_N_4_*) is unstable, and a strong cathode ECL signal will be emitted when g-CN^*^ returns to the ground state of g-CN. As shown in Figure 5b, after g-CN assembly, the modified electrode has strong ECL signal in 0.1 M PBS containing 20 mM K_2_S_2_O_8_ and 0.1 M KCl. In the process of electrode modification, Nb28-C4bpα heptamer and BSA macromolecular proteins have a certain shielding effect on the signal of g-CN on the electrode surface, resulting in the reducing of ECL signal, while OTA with a small molecular weight has a slightly different effect on ECL signal. When the electrodes were directly modified with OTA-Apt-NU-1000(Zr), the ECL-RET interaction between g-CN and NU-1000(Zr) resulted in a large decrease in the ECL signal. If a certain concentration of OTA is incubated with modified electrode prior to incubation of OTA-APT-NU-1000(Zr), OTA will first occupy some the specific binding site of Nb28-C4bpα heptamer, and OTA-Apt-NU-1000(Zr) will bind to the excess site of Nb28-C4bpα heptamer on the electrode surface. As the amount of OTA-Apt-NU-1000(Zr) bound on the electrode surface is reduced, the ECL-RET between g-CN and NU-1000(Zr) is weakened, thereby reducing the ECL signal degradation.
g-CN + e^−^ → g-CN^−^(1)
S_2_O_8_^2−^ + e^−^ → SO_4_^2−^ + SO_4_^−^(2)
g-CN^−^ + SO_4_^−^ → g-CN^*^+ SO_4_^2−^(3)
g-CN^*^ → g-CN + light(4)

Because the protein on the electrode reduces the efficiency of electron transport and inhibits the ECL reaction to form the excited state g-CN^*^, when NU-1000(Zr) were assembled to the modified electrode, the ECL signal significantly decreased, indicating that the energy of the g-CN donor is transferred to the receptor NU-1000(Zr) due to the RET on the electrode.

### 3.5. Optimization of Experimental Conditions

In immuno-recognition experiments, concentrations of Nb28-C4bpα heptamer, incubation time of Nb28-C4bpα heptamer and OTA-Apt-NU-1000(Zr) directly affect the ECL reaction in the assembly process, which ultimately influence the performance of the fabricated ECL-RET immunosensor.

For highly sensitive immune recognition of OTA, the concentration of Nb28-C4bpα heptamer is a key factor affecting specificity recognition efficiency. After incubation with different concentrations of Nb28-C4bpα, the modified electrode was incubated with low concentrations of OTA and OTA-Apt-NU-1000(Zr) complexes. During the experiment, the concentrations and incubation time of OTA and OTA-Apt-NU-1000(Zr) complexes were kept consistent. When the concentration of Nb28-C4bpα was higher, the amount of OTA-Apt-NU-1000 (Zr) complex bound on the electrode surface was higher, causing stronger ECL-RET effect, leading to a decrease in ECL signal. The ECL signal reached equilibrium at 10.0 µg/mL, suggesting that Nb28-C4bpα heptamer captured on the modified electrode was saturated. Thus, the optimal concentration of Nb28-C4bpα heptamer was 10.0 mg/mL.

In addition, the incubation time of Nb28-C4bpα heptamer also had a significant impact on the performance of the ECL-RET immunosensor. As shown in Figure 6b, with the incubation time of Nb28-C4bpα heptamer increase, the ECL signal displayed a downward trend. A rough equilibrium was reached at 60 min, indicating that sufficient Nb28-C4bpα heptamer was trapped on the electrode surface. Thus, 60 min is chosen as the optimal time for the binding of the Nb28-C4bpα heptamer.

Subsequently, the binding time of OTA-Apt-NU-1000(Zr) also has a significant impact on the performance of the ECL-RET immunosensor. In Figure 6c, the influence of the binding time of OTA-Apt-NU-1000(Zr) is presented. Results showed that the intensity of ECL decreased significantly with increasing incubation time of OTA-Apt-NU-1000(Zr), and reached a stable level after the continuous increase of OTA-Apt-NU-1000(Zr) incubation duration for 60 min. Thus, 60 min is selected as an appropriate incubation time in the experiment.

### 3.6. OTA Detection Performance of ECL-RET Immunosensor

Under optimized conditions, the quantitative detection of OTA by ECL-RET immunosensor is evaluated. In Figure 7a, when the OTA concentration increases from 0.1 pg/mL to 500 ng/mL, the high concentration of OTA caused a decrease in the binding amount of the complex OTA-Apt-NU-1000(Zr) on the electrode surface, and weakened the efficiency of ECL-RET, thus the ECL intensity was high. The ECL signal has a good linear relationship with the logarithm of OTA concentration. The linear equation is I = 1277.65 LogC_OTA_ (ng/mL) + 6345.32 (R^2^ = 0.9904), and the detection limit is 33 fg/mL (S/N = 3). In addition, this ECL-RET immunosensor was comparable to the previously reported ECL sensor, as shown in Table 1. In comparison with literatures, our constructed ECL-RET immunosensor has a wider linear range and a lower limit of detection. The high sensitivity of the proposed ECL-RET immunosensor can be attributed to the high RET efficiency between g-CN and NU-1000(Zr) and nanobody heptamer with high affinity. Nb28-C4bpα heptamer can offer more active binding sites for OTA than nanobody monomer Nb28. Compared with the shielding effect of small molecules OTA, ECL signal changes by ECL-RET between g-CN and NU-1000(Zr) are more obvious and sensitive.

### 3.7. Selectivity, Stability, and Reproducibility of the ECL Immunosensor

The specificity of the proposed ECL-RET immunosensor was investigated using the same concentration of other mycotoxins (10 ng/mL). With OTA as the control, OTB, OTC, AFB1, FB1, ZEN, and DON were incubated on the prepared ECL-RET immunosensor under the same conditions, respectively. Then, the changes in ECL response were recorded. As depicted in Figure 8a, we can clearly observe that only OTA can cause a significant reduction in ECL signal compared to the blank value. After other mycotoxins were incubated on the immunosensor and then modified with OTA-Apt-NU-1000(Zr), the change of ECL intensity distinctly decreased. The result showed that the binding rate of other toxins to Nb28-C4bpα was very low, so a large amount of Nb28-C4bpα can bind with OTA-APT-NU-1000(Zr), resulting in almost the same degree of quenching in ECL donors. Thus, the proposed ECL-RET immunosensor has good selectivity. Furthermore, we investigated the stability of constructed ECL-RET immunosensor constructed in 5 consecutive times cycles under optimal conditions at 0.0001, 0.1, and 10 ng/mL of OTA (Figure 8b). As exhibited in Figure 8b (inset), the modified electrodes containing 500 ng/mL OTA obtained a relatively stable ECL curve by 14 consecutive cycles. The RSD of ECL response was only 6.176%, illustrating that the ECL-RET immunosensor had good stability for OTA detection.

### 3.8. Spiked Sample Analysis

Recovery experiments were conducted to validate the proposed ECL-RET immunosensor in coffee samples. First, 4 mg/mL coffee powder was suspended in water and sonicated for 30 min, and the standard concentration of OTA dissolved in 0.01 M PBS solution was added to the coffee suspension, respectively. Finally, the three samples (spiked with 0.001, 1, and 100 ng/mL OTA) were centrifuged, and the concentration of OTA was assessed in the supernatants. In Table 2, the sample recovery rates of different OTA spiked concentrations are 97.486%, 100.603%, and 95.784%, respectively. The RSD does not exceed 4.214%. The results reveal that the ECL-RET immunosensor can detect coffee samples with high accuracy, which provides a new way for the detection of OTA content in coffee.

## 4. Conclusions

In general, we presented a signal amplification ECL-RET immunosensor for ultra-sensitive OTA detection using g-CN, NU-1000(Zr), and Nb28-C4bpα heptamer. In this work, the prepared Nb28-C4bpα heptamer can provide more specific capturing sites for OTA and OTA-Apt-NU-1000(Zr) than Nb28 monomers because of its high affinity. In addition, the RET between g-CN and NU-1000(Zr) can effectively display the change of ECL intensity, making up for the shortcoming that small molecules OTA have little influence on ECL signal. Thus, the synergistic effect between Nb28-C4bpα heptamer and the RET-based g-CN and NU-1000(Zr) can greatly improve the sensitivity of ECL immunosensor. The constructed immunosensor has a low detection limit and wide linear range, and exhibited good repeatability, high stability, and specificity. Finally, the developed immunosensor can successfully be used to detect the content of OTA in coffee, indicating that the method has broad application prospect in OTA content in different samples.

## Data Availability

The data are available from the corresponding author.

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
