# Peer review of "A Novel Electrochemiluminescence Immunosensor Based on Resonance Energy Transfer between g-CN and NU-1000(Zr) for Ultrasensitive Detection of Ochratoxin A in Coffee"

_foods, 2023, doi:10.3390/foods12040707_

Round 1

Reviewer 1 Report

The reported work titled as “A novel electrochemiluminescence immunosensor based on resonance energy transfer between g-CN and NU-1000(Zr) for ultrasensitive detection of ochratoxin A in coffee”, does not meet the necessary requirements to recommend its consideration for Foods in this state. Although, in general, the work could be interesting for the readers, there are many confusing issues that need to be improved. Some of the most relevant are the following:

In this work, looks like that both (nanobodies and aptamers) are used as biorecognition elements. This issue must be clarified. Which is the function of OTA aptamer in the complex defined as OTA-Apt-NU-1000(Zr)? OTA is permanently bound to the aptamer? Or as usual, the OTA molecules can be detached from the aptamers? If aptamer is used to present OTA molecules for its binding to the nanobodies, this fact has to be demonstrated since aptamers usually surround the molecule. The clarification of this aspect is a priority to understand how the system work.

Furthermore, highly relevant information as the nucleotides sequence of the aptamer must be displayed in the reagents section.

The scheme 1 should be redesigned to a better understanding. For instance, why NU-1000(Zr) is represented as a spherical nanoparticle?

The following paragraph from line 208 is not clear at all and it is incongruent with figure 5b and the supposed working mechanism of the immunosensor: “However, the ECL intensity was drastically reduced when Nb28-C4bp heptamer and BSA were fixed on the electrode. Because the protein on the electrode reduces the efficiency of electron transport and inhibits the ECL reaction to form the excited state g-CN*, however, when NU-1000(Zr) were assembled to the modified electrode, the ECL signal significantly decreased, indicates that the energy of the g-CN donor is transferred to the receptor NU-1000(Zr) due to the RET on the electrode and NU-1000(Zr) has good quenching effect”.

The following sentence is not clear at all: “As seen in figure 6a, with increase of Nb28-C4bp heptamer concentration, the ECL signal of the ECL-RET immunosensor decreased gradually, indicating that the identification and capturing of OTA antigens increased”. I suppose the signal is decreased by the presence of higher concentration of Nb28-C4bp as expressed in the figure caption. The recognition of the analyte could produce an enhanced decrease of the signal. Is it the case in this figure?

Also, the sentence: “The maximum ECL signal is achieved by suggesting 10.0 mg/mL of Nb28-C4bp heptamer for target recognition” is not appropriate.

Also this sentence is confuse: “In Figure 7a, when the OTA concentration increases from 0.1 pg/mL to 500 ng/mL, the binding amount of OTA on the surface of OTA-Apt-NU-1000(Zr) decreases, and the efficiency of ECL-RET weakens, so the ECL intensity is high”. I suppose that higher OTA concentration in the sample prevents the binding of the complex OTA-Apt-NU-1000(Zr) to the electrode surface via Nb28-C4, and hence, lower efficiency of ECL-RET.

 One of the most interesting features of immunosensors are the low analysis times. However, in this work, the analysis time is more than three hours and a half, regardless of the time spent in binding the antibody onto the electrode surface. This fact nullifies one of the clear advantages of immunosensors. This issue or an explanation has to be commented in the text.

In the stability studies and figure 8b, it is the same electrode used for more than one measurement (“consecutive time cycles”)? In case, how is the regeneration of the surface.

The epigraph 3.8 is written as “real sample analysis”. However, OTA standard solutions were added to the sample. Then, it has to be written as spiked sample analysis. Furthermore, a soluble coffee powder was analysed, but the OTA concentration was added  no to the powder but to the suspension. To mimic a real analysis of samples, the OTA had to be added to the coffee powder, dried ant etc., before the extraction and analysis.

In table 2, there are some mistakes in the recoveries of 0.001 concentration.

Author Response

To: REVIEWER 1 REPORT

Question 1In this work, looks like that both (nanobodies and aptamers) are used as biorecognition elements. This issue must be clarified. Which is the function of OTA aptamer in the complex defined as OTA-Apt-NU-1000(Zr)? OTA is permanently bound to the aptamer? Or as usual, the OTA molecules? If aptamer is used to present OTA molecules for its binding to the nanobodies, this fact has to be demonstrated since aptamers usually surround the molecule. The clarification of this aspect is a priority to understand how the system work.

Answer:According to reviewer’s advice, indeed, nanobodies and aptamers are both used as biorecognition elements in this work, and we have added descriptions in revised manuscript, “Therefore, we attempted to construct ECL-RET immunosensor using Nb28-C4bpa heptamer and OTA aptamer with high affinity for OTA molec  ules as recognition probes.” The function of OTA aptamer is biorecognition in the complex of OTA-Apt-NU-1000(Zr). In addition, OTA is permanently bound to the aptamer in OTA-Apt-NU-1000(Zr). In this research, first, rod-like NU-1000(Zr) nanoparticles with the high surface area bind large amounts of OTA molecules through the interaction between OTA and OTA aptamer as recognition element(defined as OTA-Apt-NU-1000(Zr)). Then, OTA Nbs polymer Nb28-C4bpa was applied to ECL-RET sensor as recognition elements. In the test, OTA in the sample binds to Nb28-C4bpa on the surface of the ECL-RET immunosensor, and OTA-Apt-NU-1000(Zr) combined with its remaining sites. As OTA concentration increases, the amount of OTA-Apt-NU-1000(Zr) fixed on the electrode surface decreases. Therefore, RET between g-CN and NU-1000(Zr) is attenuated, leading to an increase in ECL response. The above principle of this system is clarified in revised manuscript. Furthermore, we also compare ECL performance of different modified electrode with only OTA or OTA-Apt-NU-1000(Zr) with OTA +OTA-Apt-NU-1000(Zr) in Figure 5b. The results show that ECL response is different, suggesting that OTA in OTA-Apt-NU-1000(Zr) complex can bind to the nanobodies to some extent, although aptamers usually surround the OTA molecule.

Question 2Furthermore, highly relevant information as the nucleotides sequence of the aptamer must be displayed in the reagents section.

Answer:According to reviewer’ s advice, we have added the nucleotides sequence of the aptamer in the reagents section, the OTA aptamer with -NH2 modification (5’-NH2GATCGGGTG TGGGTGGCGTAAAGGGAGCATCGGACA-3’).

Question 3The scheme 1 should be redesigned to a better understanding. For instance, why NU-1000(Zr) is represented as a spherical nanoparticle?

Answer:According to reviewer’ s advice, we have checked and revised the scheme 1. Based on the results of TEM characterization, the NU-1000(Zr) is rod-like, and we have modified the shape of the NU-1000(Zr) to help understanding in the scheme 1.

Question 4The following paragraph from line 208 is not clear at all and it is incongruent with figure 5b and the supposed working mechanism of the immunosensor: “However, the ECL intensity was drastically reduced when Nb28-C4bpa heptamer and BSA were fixed on the electrode. Because the protein on the electrode reduces the efficiency of electron transport and inhibits the ECL reaction to form the excited state g-CN*, however, when NU-1000(Zr) were assembled to the modified electrode, the ECL signal significantly decreased, indicates that the energy of the g-CN donor is transferred to the receptor NU-1000(Zr) due to the RET on the electrode and NU-1000(Zr) has good quenching effect”.

Answer: In the process of electrode modification, Nb28-C4bpa heptamer and BSA macromolecular proteins have a certain shielding effect on the signal of g-CN on the electrode surface, resulting in the reducing of ECL signal, while OTA with a small molecular weight has a slightly effect on ECL signal. When the electrodes were directly modified with OTA-Apt-NU-1000(Zr), the ECL-RET interaction between g-CN and NU-1000(Zr) resulted in a large decrease in the ECL signal. If a certain concentration of OTA is incubated with modified electrode prior to incubation of OTA-APT-NU-1000(Zr), OTA will first occupy some the specific binding site of Nb28-C4bpa heptamer, and OTA-Apt-NU-1000(Zr) will bind to the excess site of Nb28-C4bpa heptamer on the electrode surface. As the amount of OTA-Apt-NU-1000(Zr) bound on the electrode surface reduced, the ECL-RET between g-CN and NU-1000(Zr) is weakened, thereby reducing the ECL signal degradation. The results in figure 5b was consistent with this working mechanism.

Question 5The following sentence is not clear at all: “As seen in figure 6a, with increase of Nb28-C4bpa heptamer concentration, the ECL signal of the ECL-RET immunosensor decreased gradually, indicating that the identification and capturing of OTA antigens increased”. I suppose the signal is decreased by the presence of higher concentration of Nb28-C4bpa as expressed in the figure caption. The recognition of the analyte could produce an enhanced decrease of the signal. Is it the case in this figure?Also, the sentence: “The maximum ECL signal is achieved by suggesting 10.0 mg/mL of Nb28-C4bpa heptamer for target recognition” is not appropriate.

Answer:According to reviewer’ s advice, we revised the article and added explanations for the experimental procedure and results section. In figure 6a, after incubation with different concentrations of Nb28-C4bpa, the modified electrode was incubated with low concentrations of OTA and OTA-Apt-NU-1000(Zr) complexes. During the experiment, the concentrations and incubation time of OTA and OTA-Apt-NU-1000(Zr) complexes were kept consistent. When the concentration of Nb28-C4bpa was higher, the amount of OTA-Apt-NU-1000(Zr) complex bound on the electrode surface was higher, causing stronger ECL-RET, leading to a decrease in ECL signal. The ECL signal reached equilibrium at 10.0 mg/mL, suggesting that Nb28-C4bpaa heptamer captured on the modified electrode was saturated. Thus, the optimal concentration of Nb28-C4bpa heptamer was 10.0 mg/mL.

Question 6Also this sentence is confuse: “In Figure 7a, when the OTA concentration increases from 0.1 pg/mL to 500 ng/mL, the binding amount of OTA on the surface of OTA-Apt-NU-1000(Zr) decreases, and the efficiency of ECL-RET weakens, so the ECL intensity is high”. I suppose that higher OTA concentration in the sample prevents the binding of the complex OTA-Apt-NU-1000(Zr) to the electrode surface via Nb28-C4, and hence, lower efficiency of ECL-RET.

Answer:According to reviewer’ s advice, we have checked and revised the article. Indeed, the high concentration of OTA caused a decrease in the binding amount of OTA-Apt-NU-1000(Zr) the complex on the electrode surface, and weakened the efficiency of ECL-RET, thus the ECL intensity was high.

Question 7One of the most interesting features of immunosensors are the low analysis times. However, in this work, the analysis time is more than three hours and a half, regardless of the time spent in binding the antibody onto the electrode surface. This fact nullifies one of the clear advantages of immunosensors. This issue or an explanation has to be commented in the text.

Answer:According to reviewer’ s advice, the analysis time of immunosensors is more than three hours and a half in this work, but detection time is quick, only several minutes. In addition, developed immunosensors have other advantages, such as high sensitivity and stability, low cost and background signal. We will continue to reduce analysis times in next work.

Question 8In the stability studies and figure 8b, it is the same electrode used for more than one measurement (“consecutive time cycles”)? In case, how is the regeneration of the surface.

Answer:We investigated the stability of constructed ECL-RET immunosensor constructed in 5 consecutive times cycles under optimal conditions at 0.0001, 0.1, and 10 ng/mL of OTA (Figure 8b). As exhibited in Figure 8b (inset), the modified electrodes containing 500 ng/mL OTA obtained a relatively stable ECL curve by 14 consecutive cycles. The RSD of ECL response was only 6.176%, illustrating that the ECL-RET immunosensor had good stability for OTA detection. Regeneration of the surface is based on a g-CN-K2S2O8 system in Scheme 2. In this reaction system, firstly, g-CN·-, SO42-, and SO4·- are generated from g-CN and S2O82- respectively, and g-CN·- reacts with SO4·- to form the excited state g-CN*. The excited state g-CN* is unstable, and a strong cathode ECL signal will be emitted when g-CN* returns to the ground state of g-CN. When consecutive time cycles of modified electrode were conducted, ECL generation is repeated according to the above reaction mechanism.

Question 9The epigraph 3.8 is written as “real sample analysis”. However, OTA standard solutions were added to the sample. Then, it has to be written as spiked sample analysis. Furthermore, a soluble coffee powder was analysed, but the OTA concentration was added no to the powder but to the suspension. To mimic a real analysis of samples, the OTA had to be added to the coffee powder, dried ant etc., before the extraction and analysis.

Answer:According to reviewer’ s advice, we have checked the article and changed the epigraph 3.8. In the process of preparing the sample solution, we referred to the standard method of adding OTA before obtaining the supernatant by centrifugation, so that OTA was mixed with the coffee suspension thoroughly. This was in accordance with the preparation process of the standard addition method.

Question 10In table 2, there are some mistakes in the recoveries of 0.001 concentration.

Answer:We calculated the “detected” and “recovery” based on the original data. In order to maintain the uniformity of the table, all three decimal places were retained in the table. Therefore, the “detected” was all taken as 0.001, but the “recovery” was slightly different.

Reviewer 2 Report

Some corrections and suggestions to improve the manuscript:

Line 42-43: with IARC it is enough

Line 66: Define CdS QDs, and CdTe

Line 158: Some contaminating bands are observed in the lanes of the gel, for the monomer and heptamer. They are bands that are not of the expected weight. If I could have a silver stain, to see that the proteins are pure and if not; try to argue why those proteins that can be seen in the gel do not interfere in the experiment.

Line 273: OTB, OTC, AFB1, FB1, ZEN and DON. Although they are defined in the supplementary material, also define them here. and Also in the figure where they are mentioned, write what they mean in the figure caption.

Fix some typos in the text.

Author Response

To: REVIEWER 2 REPORT

Question 1: Line 42-43: with IARC it is enough

Answer:According to reviewer’s advice, we have checked and revised the article.

Question 2: Line 66: Define CdS QDs, and CdTe

Answer: According to reviewer’s advice, we have defined CdS QDs (cadmium sulfide semiconductor quantum dots) in article. CdTe is a compound and not an abbreviation.

Question 3: Line 158: Some contaminating bands are observed in the lanes of the gel, for the monomer and heptamer. They are bands that are not of the expected weight. If I could have a silver stain, to see that the proteins are pure and if not; try to argue why those proteins that can be seen in the gel do not interfere in the experiment.

AnswerAccording to the results of SDS-PAGE, the purity of the Nb28-C4bpa heptamer was high. Although a small part of the heptamer was degraded into monomeric proteins, the amount of monomeric proteins was few, which would not affect the experimental results. In addition, there were no other proteins with higher concentrations in the bands of the heptamers, so they do not interfere with the experiments.

Question 4: Line 273: OTB, OTC, AFB1, FB1, ZEN and DON. Although they are defined in the supplementary material, also define them here. and Also in the figure where they are mentioned, write what they mean in the figure caption.

Answer:According to reviewer’ s advice, we have checked and added the definitions in “2.1. Chemical reagents and apparatus” in revised manuscript.

Question 5: Fix some typos in the text.

Answer:According to reviewer’s advice, we have checked the article and revised some typos.

Question 6: one of the referees has suggested that your manuscript should undergo extensive English revisions, please address this issue during revision.

Answer: Based on reviewer’s advice, we have checked the English in the whole manuscript.

Round 2

Reviewer 1 Report

Authors have answered correctly to the questions and comments. Besides, some relevant modifications in the text allow to better understand the work.

Author Response

Thanks for your suggestion.